# Fortifying Time Series: DTW-Certified Robust Anomaly Detection

**Shijie Liu**[1]*, **Tansu Alpcan**[1], **Christopher Leckie**[2], **Sarah Erfani**[2]
[1]Department of Electrical and Electronic Engineering
University of Melbourne, Melbourne, Australia
[2]School of Computing and Information Systems
University of Melbourne, Melbourne, Australia
*shijie3@unimelb.edu.au

## Abstract

Time-series anomaly detection is critical for ensuring safety in high-stakes applications, where robustness is a fundamental requirement rather than a mere performance metric. Addressing the vulnerability of these systems to adversarial manipulation is therefore essential. Existing defenses are largely heuristic or provide certified robustness only under $\ell_p$-norm constraints, which are incompatible with time-series data. In particular, $\ell_p$-norm fails to capture the intrinsic temporal structure in time series, causing small temporal distortions to significantly alter the $\ell_p$-norm measures. Instead, the similarity metric *Dynamic Time Warping* (DTW) is more suitable and widely adopted in the time-series domain, as DTW accounts for temporal alignment and remains robust to temporal variations. To date, however, there has been no certifiable robustness result in this metric that provides guarantees. In this work, we introduce the first *DTW-certified robust defense* in time-series anomaly detection by adapting the randomized smoothing paradigm. We develop this certificate by bridging the $\ell_p$-norm to DTW distance through a lower-bound transformation. Extensive experiments across various datasets and models validate the effectiveness and practicality of our theoretical approach. Results demonstrate significantly improved performance, e.g., up to 18.7% in F1-score under DTW-based adversarial attacks compared to traditional certified models.

## 1 Introduction

In recent years, significant research has advanced the study of adversarial attacks and certified defenses for machine learning systems. Despite the considerable progress in adversarial robustness across various domains [42, 2, 45, 9, 12, 3], robustness in *time-series anomaly detection* remains comparatively underexplored. As a core component of many safety-critical systems—including healthcare [25, 46, 21], finance [41, 22, 64], and mobile networks [56, 70, 35]— anomaly detectors are essential for identifying abnormal behavior in preventing failures or hazards. Robustness in this context is not merely a model performance concern but a core requirement for operational reliability. Recent work has revealed that time-series anomaly detectors are susceptible to adversarial attacks tailored to the characteristics of time-series data [6, 5], underscoring the urgent need for ensuring robustness in this domain.

Adversaries can manipulate detection outcomes by introducing subtle yet strategically crafted perturbations into anomaly time-series data to evade detection [61, 29, 74, 69]. Traditional adversarial threat models typically restrict perturbations to bounded $\ell_p$-norms, widely effective in image [1, 31] and text domains [63, 76, 57] due to the alignment with semantic preservation in such data types. However, time-series data exhibit an inherent *temporal structure* that challenges the assumption of

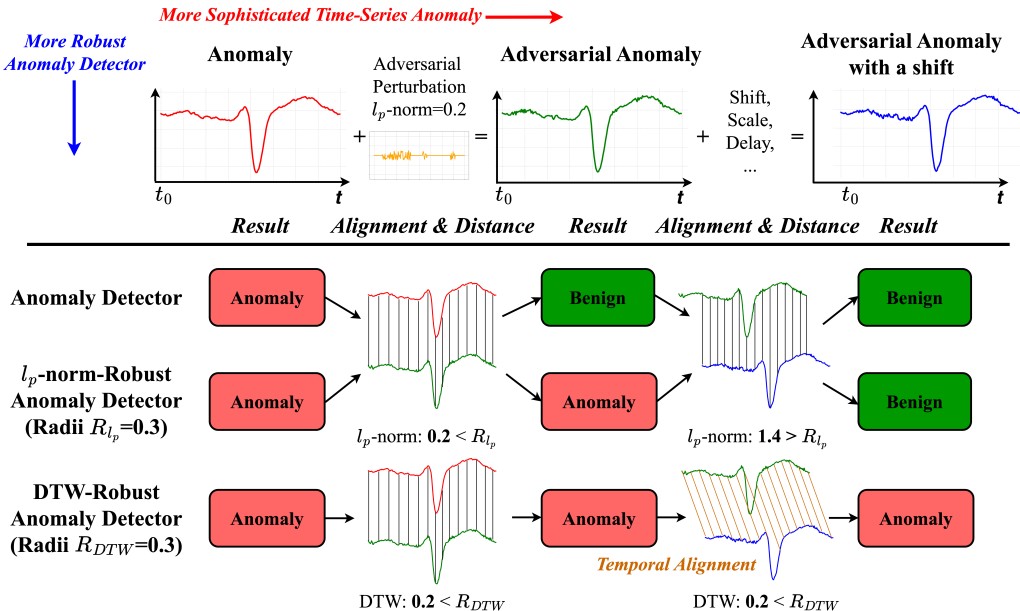

Figure 1: Comparison of standard, $\ell_p$-norm-robust, and DTW-robust anomaly detectors under adversarial perturbations. DTW facilitates optimal temporal alignment, offering a more meaningful similarity measure for time-series data, thereby ensuring more comprehensive robustness guarantees against adversarial examples.

$\ell_p$-norms. As illustrated in Figure 1, small temporal transformations such as shifts or rescaling can significantly inflate $\ell_p$-norm distance (e.g., from $0.2$ to $1.4$), despite there being no change to the underlying semantics (e.g., green and blue represent the identical time series). This mismatch renders $\ell_p$-norms inadequate for measuring meaningful similarity in time-series data, limiting their utility for robustness measurement.

Addressing this limitation, we advocate for the use of *Dynamic Time Warping (DTW)* distance, a commonly used similarity metric specifically designed for time-series data. As shown in the bottom right of Figure 1, DTW accommodates *temporal alignments*, which effectively handles temporal variations such as shifts, stretching, and compression. This alignment flexibility preserves structural similarities better than $\ell_p$-norms, consistently demonstrating superior performance for diverse time-series tasks [24, 37, 19]. As a result, a DTW-robust detector exhibits stable distance measurements under temporal variations (e.g., consistently $0.2$ as in Figure 1), offering more reliable robustness guarantees compared to $\ell_p$-norm. The need for developing defenses under the DTW distance is further emphasized by recent demonstrations of DTW-based adversarial attacks [6, 5], for which no certified defenses currently exist.

In response to adversarial attacks, various defensive strategies have been proposed [38, 11, 20]. Although these empirical defenses provide some resilience, adaptive attackers can often bypass them [10, 65, 75]. *Certified defenses* [32, 18, 55], in contrast, guarantee theoretical robustness against worst-case adversarial scenarios, making them particularly appealing for safety-critical applications. While significant progress has been made in certified robustness under $\ell_p$-norm constraints [32, 55, 18, 54, 33, 72], its adaptation to time-series data—under the proper DTW constraints—remains unexplored.

In this paper, we propose the first *DTW-certified robustness* framework in time-series anomaly detection by adapting the randomized smoothing approach [18]. Our approach establishes a novel certification method by bridging $\ell_p$-norm guarantees to the DTW distance through a lower-bound transformation. By leveraging the Keogh Lower Bound [28], we are able to derive a closed-form expression of the DTW-certified radius for a smoothed model. The resulting framework is model-agnostic and readily applicable to any pre-trained anomaly detector. Extensive evaluations on real-world datasets and a variety of detection architectures highlight the broad applicability and effectiveness of our method, demonstrating clear advantages over traditional $\ell_p$-norm-certified defenses,

e.g., under DTW-based adversarial attacks, our method achieves up to an 18.7% improvement in F1-score.

Our key contributions include:

- We introduce the first theoretical framework that provides certified robustness in DTW distance, addressing an essential yet unexplored gap in time-series anomaly detection.
- We present a generalizable defense mechanism that seamlessly integrates DTW certification with any anomaly detection models, significantly enhancing robustness in practice.
- We provide comprehensive experimental evaluations demonstrating the practical efficacy of our DTW-certified robustness approach across diverse scenarios.

## 2   Related Work

**Time-series Anomaly Detection**  Time-series data, consisting of data points sequentially indexed over time, is prevalent across various domains. Detecting anomalies within time-series data is of significant importance [40, 49, 16, 7, 25, 46, 21], as anomalies often indicate novel, unexpected, or potentially critical events. Recent advances in deep learning have significantly improved detection by enabling models to capture complex temporal and inter-metric dependencies. Modern *deep anomaly detectors* [62, 13, 36] have shown strong performance across a range of time-series tasks. However, they also share the same vulnerabilities to adversarial attacks as other machine learning models.

**Adversarial Attacks**  Adversarial attacks refer to deliberate perturbations introduced into input data to intentionally mislead machine learning models into making incorrect predictions. Typically, these perturbations are minimal in terms of the $\ell_p$-norm, ensuring the semantic consistency and being imperceptible to humans. Such vulnerabilities have been widely demonstrated across various deep learning models [1, 31, 63, 76, 57], including anomaly detection tasks [61, 29, 74, 69].

However, the $\ell_p$-norm is inadequate for measuring differences in *time-series data*, as it fails to account for the underlying temporal structure. Recent studies [6, 5] have addressed these issues by adopting the DTW distance, a widely recognized measure suitable for time-series analysis, to construct adversarial examples. These studies highlight that DTW-based adversarial attacks are more effective, as the set of permissible perturbations under DTW forms a superset of those constrained by an equivalent $\ell_p$-norm. Additionally, they demonstrate that the defensive strategies designed to counter $\ell_p$-norm adversarial attacks exhibit limited effectiveness against DTW-based attacks [39], highlighting the need for dedicated defenses under the DTW threat model.

**Certified Robustness**  Prior defenses such as adversarial training [39], defensive distillation [44], and data purification [66] offer empirical robustness, but are often circumvented by adaptive adversaries [10, 65, 75]. In contrast, *certified defenses* have gained significant attention for providing formal, provable guarantees against all possible attacks within a perturbation bound [32, 18, 53]. In the context of time-series anomaly detection, certified defenses against $\ell_p$-norm attacks have been initially considered in [23, 8], through direct application of randomized smoothing [18]. However, as discussed, the resulting $\ell_p$-norm certificate is inadequate for time-series data and remains vulnerable to DTW-based attacks. Existing defences [6] against DTW-based attacks have been limited to empirical without any certification. This work introduces the first certified defense against DTW-based adversarial attacks.

## 3   DTW-Certified Defense in Time-Series Anomaly Detection

### 3.1   Problem Setup

**Time-Series Anomaly Detector**  We define the space of time-series signals as $\mathcal{X} = \mathbb{R}^{L \times C}$, where $L$ represents the signal length and $C$ denotes the number of channels. Following the common framework for time-series anomaly detection [71, 67, 51], we consider a detector $d : \mathbb{R}^{T \times C} \rightarrow \mathcal{Y} = \{0, 1\}$ that operates on a sliding window of size $T \leq L$. Given an input sequence $x \in \mathbb{R}^{T \times C}$, the detector computes an *anomaly score* $f(x) \in \mathbb{R}$, which quantifies the likelihood of $x$ being an anomaly, and makes the detection decision via comparing $f(x)$ against the anomaly threshold $\gamma$:

$$d(x) = \begin{cases} 1, & f(x) > \gamma \ , \\ 0, & f(x) \leq \gamma \ , \end{cases} \tag{1}$$

where $y = 1$ indicates an anomalous instance, and $y = 0$ denotes a benign instance.

**Distance Metrics** The difference between two time-series $x$ and $x'$ can be naively measured by the $\ell_p$-*norm distance* as

$$\|x - x'\|_p = \left( \sum_{i=1}^{T} |x_i - x_i'|^p \right)^{1/p}, \tag{2}$$

where $x_i, x_i'$ represent the $i$-th element in $x, x'$. However, such a measurement fails to capture the temporal structure of time-series data. Therefore, we consider the *Dynamic Time Warping (DTW) distance*, which resolves the issues by finding the optimal temporal alignment that minimizes the total distance between aligned time-series. Formally, the DTW distance of norm order $p$ is defined as:

$$DTW_p(x, x') = \min_{\pi \in \mathcal{A}(x,x')} \left( \sum_{(i,j) \in \pi} |x_i - x_j'|^p \right)^{1/p} \tag{3}$$

where $\pi$ represents an *alignment path* of length $T$ as a sequence of $T$ index pairs $[(i_1, j_1), \cdots, (i_T, j_T)]$ and $\mathcal{A}(x, x')$ is the set of all admissible paths. An admissible path should satisfy the following conditions: 1) Matched ends, as $\pi_1 = (1, 1)$ and $\pi_T = (T, T)$, and 2) Monotonically increasing and each time series index should appear at least once, as $i_{k-1} \leq i_k \leq i_{k-1} + 1$ and $j_{k-1} \leq j_k \leq j_{k-1} + 1$. We adopt $p = 2$ as the default norm for DTW in the main text for clarity of exposition; however, the proposed approach generalizes readily to arbitrary norm orders $p$ with minimal modification as detailed in Appendix C.

**Threat Model** We assume a strong adversary with white-box access to the anomaly detector $d$, meaning the attacker has full knowledge of the detector and unlimited computational power. Given an input $x$ classified by the detector as $y = f(x)$, the attacker seeks an alternative input $x'$ to perform either an *evasion attack*—suppressing the detection of an actual anomaly, or an *availability attack*—inducing a false alarm on benign input, such that $d(x') \neq d(x)$. To preserve the semantics of the original anomaly $x$, the perturbation in $x'$ must be constrained within a DTW distance $e$ as $DTW(x, x') < e$.

**Certified Defense Goal** The anomaly detector $d$ is said to provide certified defense at input $x$ of DTW radius $e$, if there exist no $x' \in \{x' \mid DTW(x, x') < e\}$ such that $d(x') \neq d(x)$ with probability at least $1 - \alpha$.

### 3.2 Theoretical Analysis of DTW-Certified Robustness

In this section, we first review the core components of our approach: the randomized smoothing framework [18, 15] and the DTW lower bound [28]. We then present Lemma 3.2, which establishes a formal link between $\ell_p$-norm distances and the DTW lower bound. Building on this connection, we introduce the main theoretical result Theorem 3.3, which derives a DTW robustness certificate from a smoothed model via the Keogh Lower Bound.

**$\ell_p$-norm Certificate via Randomized Smoothing** Randomized smoothing [17] constructs a *smoothed function* by taking the Gaussian expectation of a base function $f$ (we defer the details in Appendix A). However, in time-series anomaly detection, the base function $f(x)$—which outputs an anomaly score for a time series $x$—is typically unbounded and may exhibit high variance. As a result, estimating the Gaussian mean can lead to loose and unreliable robustness bounds.

To address this, we adopt the *percentile smoothing* approach [14], which bounds the $p$-*th percentiles* of the base function outputs instead of the mean. Such a smoothing method is more robust to outliers and variance in the output distribution. We construct the *smoothed anomaly score function* $h_p(x) : \mathcal{X} \to \mathbb{R}$ of the anomaly score function $f$, as

$$h_p(x) = \sup\{u \in \mathbb{R} \mid \mathbb{P}_{\eta \sim N(0, \sigma^2 I)}[f(x + \eta) \leq u] \leq p\}. \tag{4}$$

The $h_p$ does not admit a closed form, its value can be bounded by Monte Carlo sampling as outlined in Section 3.3. With the percentile smoothed function, the anomaly score $h_p(x')$ of the adversarial input $x'$ can be certifiably bounded by $h(x)$, as

**Lemma 3.1.** *A percentile smoothed function $h_p$ can be bounded as*

$$h_{\underline{p}}(x) \leq h_p(x') \leq h_{\overline{p}}(x) \quad \forall x' \in \{x' \mid \|x - x'\|_2 \leq r\}, \tag{5}$$

*where $\underline{p} = \Phi(\Phi^{-1}(p) - \frac{r}{\sigma})$ and $\overline{p} = \Phi(\Phi^{-1}(p) + \frac{r}{\sigma})$, with $\Phi$ being the standard Gaussian CDF.*

**Lower Bound of DTW**    The exact computation of DTW is typically expensive and slow, i.e., quadratic time and space complexity. To address this, various lower bounds have been proposed to approximate DTW efficiently. One of the widely used bounds is the Keogh Lower Bound [28, 47] $LB\_Keogh(x, x')$, which is calculated by defining two new time series, upper $U$ and lower $L$ envelopes. For each time step $i$ and channel $k$, the envelopes are defined as:

$$U_{i,k} = \max(x_{i-w,k} : x_{i+w,k})$$
$$L_{i,k} = \min(x_{i-w,k} : x_{i+w,k})$$

(6)

where the $w : 1 \le w \le T$ is the DTW wrapping window size (Sakoe–Chiba band) [52] that constrains only $x_i$ and $x'_j$ within the window can be aligned. The $LB\_Keogh(x, x')$ is calculated as

$$LB\_Keogh_p(x, x') = \sqrt[p]{\sum_{i=1}^{T} \sum_{k=1}^{N} \begin{cases} (x'_{i,k} - U_{i,k})^p & \text{if } x'_{i,k} > U_{i,k}, \\ (x'_{i,k} - L_{i,k})^p & \text{if } x'_{i,k} < L_{i,k}, \\ 0 & \text{otherwise.} \end{cases}}$$

(7)

In summary, the lower bound is calculated as the sum of $\ell_p$-norm distances to the envelope of points in $x'$ that are outside the envelope of $x$.

**DTW-Certificate**    In the following, we present the theoretical foundation for deriving DTW-certified robustness, offering a robustness measure that is better aligned with the temporal nature of time-series data. By leveraging the percentile-smoothed function and the DTW lower bound, we introduce a lemma that establishes a connection between the $\ell_p$-norm certificate and a robustness certificate in DTW distance through a lower-bound transformation.

**Lemma 3.2.** *Suppose the certification of a smoothed function $h$ holds for data $x$ as $a \le h(x') \le b, \forall x' \in \{x' \mid \|x' - x\| \le r\}$. Then, the certification $a \le h(x') \le b, \forall x' \in \{x' \mid \mathrm{DTW}(x, x') \le e\}$ also holds, where $LB(x, x')$ is a strict lower bound of $\mathrm{DTW}(x, x')$ and*

$$e = \inf\{LB(x, x') \mid \|x - x'\| > r\}.$$

(8)

*Proof.* Assume for the sake of contradiction that the chosen $x'$ does not lie in the $l_2$-ball. Then we have $\|x' - x\| > r$. Since $x'$ is outside the ball, by the definition of $e$ we know that $LB(x, x') \ge e$. This contradicts $LB(x, x') < DTW(x, x') \le e$ by the definition of the set $\{x' \mid \mathrm{DTW}(x, x') \le e\}$ and $LB$ is a strict lower bound of $DTW$. Thus, we conclude that any point $x'$ with $DTW(x, x') \le e$ must satisfy $\|x' - x\| \le r$, where the certification holds. $\qquad\square$

Building upon Lemma 3.2, we present the theorem that establishes DTW-certified robustness for the smoothed function. Formally, we define the anomaly score function $f : \mathcal{X} \to \mathbb{R}$ of a time-series anomaly detector $d : \mathcal{X} \to \mathcal{Y}$, and construct a percentile smoothed version of $f$, denoted as $h_p : \mathcal{X} \to \mathbb{R}$, which serves as the new anomaly score function of $d$. The DTW-certified robustness of $d$ can then be derived through the following theorem.

**Theorem 3.3** (Robustness Certification for Time-series Anomaly Detection). *Let $f : \mathcal{X} \to \mathbb{R}$ be any deterministic or random function, and $\eta \sim \mathcal{N}(0, \sigma^2 I)$. Let the percentile smoothed function $h_p : \mathcal{X} \to \mathbb{R}$ be defined as in Equation (4). Suppose the anomaly score threshold is $\gamma$, and the following is satisfied for a testing input $x$*

$$\begin{cases} h_{\underline{p}}(x) > \gamma, & \text{if } h_p(x) > \gamma, \\ h_{\overline{p}}(x) \le \gamma, & \text{if } h_p(x) \le \gamma. \end{cases}$$

(9)

*Then $d(x') = d(x)$ is guaranteed to hold for all $\{x' : DTW(x, x') \le e\}$, where*

$$e = \begin{cases} 0, & \text{if } r \le R, \\ \sqrt{M^2 + r^2 - R^2} - M, & \text{if } r > R, \end{cases}$$

(10)

*with*

$$\Delta_i = \max(U_i - x_i, \, x_i - L_i), \quad R = \sqrt{\sum_{i=1}^{n} \|\Delta_i\|^2}, \quad M = \max_{1 \le i \le n} \|\Delta_i\|,$$

$$r = \begin{cases} \sigma\big(\Phi^{-1}(p) - \Phi^{-1}(\underline{p})\big), & \text{if } h_p(x) > \gamma, \\ \sigma\big(\Phi^{-1}(\overline{p}) - \Phi^{-1}(p)\big), & \text{if } h_p(x) \le \gamma. \end{cases}$$

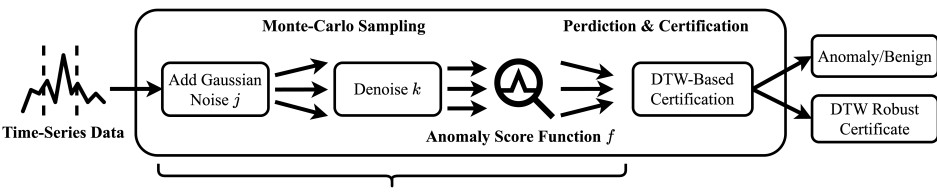

Figure 2: Construct any anomaly detector with anomaly score function $f$ as a DTW-certified detector.

*where $U$ and $L$ are envelopes in the Keogh Lower Bound with wrapping window size $w$ as specified in Equation (6).*

*Proof.* We provide a sketch of the proof below due to space constraints, and the complete proof is available in Appendix B.

We begin by showing that $d(x') = d(x)$ holds for all $x'$ satisfying $\|x - x'\| \le r$ using Lemma 3.1. The radius $r$ can be solved as $r = \sigma\left(\Phi^{-1}(p) - \Phi^{-1}(\underline{p})\right)$ or $r = \sigma\left(\Phi^{-1}(\overline{p}) - \Phi^{-1}(p)\right)$ depending on the classification outcome. Next, we invoke Lemma 3.2 to translate the certificate from $\ell_p$-norm $r$ to DTW distance $e$, defined as $e = \inf\{LB(x, x') \mid \|x - x'\| > r\}$, where $LB(x, x')$ is a strict lower bound of the DTW distance. In the final step, we instantiate $LB(x, x')$ with the Keogh Lower Bound, which satisfies the strictness condition for all $w > 0$ and $x' \ne x$, and derive the corresponding expression for $e$ by exploiting the structural properties of the upper and lower envelopes $U$ and $L$. $\square$

### 3.3 DTW-Certified Defense Implementation

**Construct smoothed detector** Given an anomaly detector with an anomaly score function $f$, we construct the percentile-smoothed anomaly score function $h_p$ by the definition of Equation (4) following the process as shown in Figure 2. Specifically, the smoothed function is composed as $h_p = j \circ k \circ f$, where the smoothing noise $\eta \sim \mathcal{N}(0, \sigma^2 I)$ injection layer $j$ generates multiple Gaussian-perturbed inputs $x + \eta$ from the original time-series $x$, and the denoising layer $k$ reduces noise variance to improve score concentration. This denoising process does not compromise the certification guarantee, as randomized smoothing is valid for any downstream pipeline [54]. For each testing input $x$, the DTW-certified anomaly detector outputs a binary decision based on anomaly score $h_p(x)$ and computes the corresponding certified DTW radius $e$ by the Theorem 3.3. This method does not require modifications to the training process and can be readily applied to pre-trained models.

**Bound $h_{\overline{p}}(x)$ and $h_{\underline{p}}(x)$** We utilize Monte-Carlo sampling to estimate and bound the upper and lower percentiles $h_{\overline{p}}(x)$ and $h_{\underline{p}}(x)$, following a similar approach as in [18, 15]. Given $n$ i.i.d. Gaussian noise samples $\{\mu_1, \cdots, \mu_n\}$, we compute anomaly scores $X_i = f(x + \mu_i)$ and sort them to obtain the empirical order statistics $-\infty = K_0 \le K_1, \cdots \le K_n \le K_{n+1} = \infty$. We aim to identify $K_{q^u}$ and $K_{q^l}$ such that $\Pr[K_{q^u} > h_{\overline{p}}(x)] > 1 - \alpha$ and $\Pr[K_{q^l} < h_{\underline{p}}(x)] > 1 - \alpha$ for a confidence level of $1 - \alpha$ (we set $\alpha = 1e{-}3$ in the experiments). The corresponding probabilities are evaluated using the binomial distribution as

$$\Pr[K_{q^u} > h_{\overline{p}}(x)] = \sum_{i=1}^{j=q^u} \binom{n}{i} (\bar{p})^i (1 - \bar{p})^{n-i} . \tag{11}$$

A similar formula applies for the lower bound. We use binary search to identify the smallest $q^u$ and largest $q^l$ that satisfy the required confidence bounds. In general, increasing the number of samples improves the estimation accuracy of the certified radius, and greater consensus aggregated predictions indicate a stronger certification.

## 4 Experiments

In our experiments, we evaluate the general applicability of the DTW-certified defense across a range of anomaly detection models and time-series datasets. We demonstrate improved robustness

compared to $\ell_p$-norm certified defenses and provide ablation studies to analyze the trade-off between detection performance and certified robustness.

**Settings**  Our empirical evaluation of the DTW-certified defense spans seven widely used benchmark datasets, including SMAP [48], MSL [27], SML [60], NIPS-TS-SWAN, NIPS-TS-CREDITCARD, NIPS-TS-WATER [30], UCR-1 ane UCR-2 [68], encompassing both univariate and multivariate time-series data. Detailed descriptions and dataset statistics are provided in Appendix D. To ensure broad applicability, we evaluate our approach using three state-of-the-art anomaly detection models: COUTA [71], TimesNet [67], and DeepSVDDTS [51]. The effectiveness is further validated through comparison with $\ell_p$-norm certified defense [18] under DTW-based adversarial attack [6].

We use the following default hyperparameters across all experiments unless otherwise specified: sequence length $T = 50$, DTW wrapping window size $w = 4$, number of noisy samples $n = 1,000$, smoothing noise level $\sigma = 0.5$ in $\mathcal{N}(0, \sigma^2 I)$, and percentile $p = 0.5$ in the percentile-smoothed function $h_p$. Additional ablation studies on the hyperparameters are available in Appendix F.

All experiments are implemented using PyTorch and executed on a Linux server equipped with Intel(R) Xeon(R) Gold 6326 CPUs and NVIDIA A100 GPUs with 80 GB of memory.

**Evaluation Metrics**  For evaluating the *detection performance*, we report the point-adjusted **F1-score** and Area Under the Receiver Operating Characteristic Curve (**ROC AUC**) following the common practice in the domain of time-series anomaly detection [4, 50, 58, 59, 34, 71, 51, 67].

To evaluate *certified robustness*, we report the **mean**, **maximum**, and **standard deviation (std.)** of the certified radii computed for all test instances. Additionally, we report the **certified proportion (prop.)**, defined as the fraction of test inputs with a non-zero certified radius $e$.

Following the notion of *certified accuracy* from the certified robustness literature [32, 18, 53], which is defined as the proportion of instances for which the model guarantees correct predictions within a specified *attack budget* as radius $t$. We extend this evaluation to the confusion matrix components by considering worst-case adversarial scenarios. Specifically, for evasion attacks, we define Certified True Positives (TP) as the count of true positive instances for which the model is provably robust within a DTW radius of $t$ as $\sum_{i=1}^{N} \mathbb{I}\{\forall x' : DTW(x_i, x') \leq t : f(x') = 1, y_i = 1\}$. Similarly, for availability attacks, we define the Certified True Negatives (TN) as the number of benign instances that remain correctly classified under all perturbations within the DTW radius $t$ as $\sum_{i=1}^{N} \mathbb{I}\{\forall x' : DTW(x_i, x') \leq t : f(x') = 0, y_i = 0\}$. We construct the corresponding certified confusion matrix as detailed in Appendix E, and derive the certified metrics, **certified accuracy** and **certified F1-score** that represent the guaranteed performance under bounded attack.

## 4.1  Results

**The DTW-certified defense is broadly applicable, though the certified robustness performance varies across datasets and models.**  We evaluate the applicability of our DTW-certified defense across benchmark datasets and anomaly detection models. As shown in Table 1, our approach generally achieves strong certified robustness with minimal trade-offs in detection performance. For instance, on the NIPS-TS-WATER using DeepSVDDTS, our method certifies 99.46% of test inputs with an average certified robust DTW-radius of 0.189, without any degradation in F1-score and ROC AUC. Additionally, DeepSVDDTS often achieves the strongest performance, which we attribute to its superior handling of noisy data. However, we observe weaker robustness on certain datasets, such as SMAP and NIPS-TS-SWAN, due to their high channel dimensionality and greater data variance, which reduces the tightness of the lower-bound estimation and thus limits certifiable robustness.

Figure 3 presents the certified F1-score and certified accuracy of the COUTA model on the MSL and SMAP datasets under evasion and availability attacks. The x-axis denotes the attack budget radius $t$, while the y-axis shows the corresponding certified metrics. These curves represent lower bounds on model performance under the worst-case adversarial perturbations constrained by $DTW(x, x') \leq t$, as guaranteed by our DTW-certified defense. With appropriately chosen hyperparameters (e.g., $\sigma = 1.0$), the defense exhibits strong certified robustness. For instance, on the SMAP dataset, it maintains a certified F1-score of approximately 0.5 under an evasion attack with a budget of $t = 0.2$.

**Improved performance over $\ell_p$-norm certified defense.**  Table 2 evaluates the effectiveness of our DTW-certified defense under strong DTW-based adversarial attacks [6]. The adversary is granted a

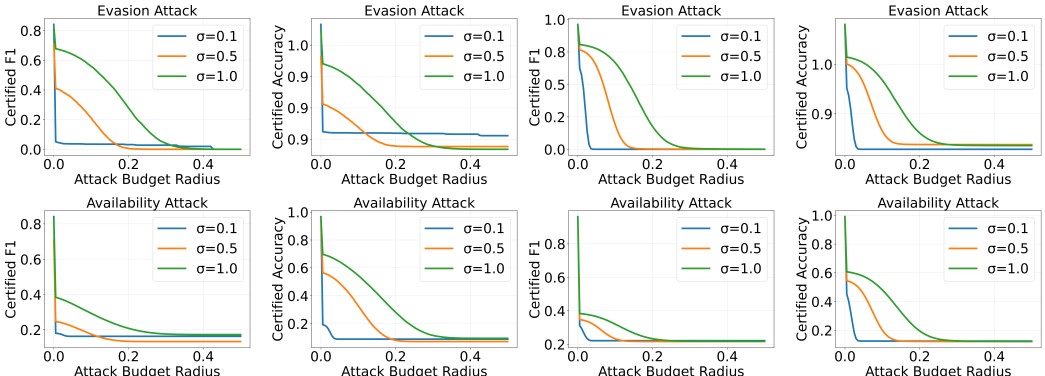

(a) Certified metrics evaluated on the MSL dataset. (b) Certified metrics evaluated on the SMAP dataset.

Figure 3: Certified accuracy and certified F1-score as functions of the DTW perturbation threshold $t \in [0.0, 0.5]$ under evasion or availability attack. Results are reported for the COUTA model on the MSL (a) and SMAP (b) datasets across varying values of the hyperparameter $\sigma$.

| Dataset | Model | Base Model | | DTW-Certified Defense Model | | | | | |
|---|---|---|---|---|---|---|---|---|---|
| | | Detection Performance | | Detection Performance | | Certified Robustness (Radii Statistics) | | | |
| | | F1-score | ROC AUC | F1-score | ROC AUC | Radii Mean | Radii Max | Radii Std. | Certified Prop. |
| SMAP | COUTA | 0.794 | 0.958 | 0.961 | 0.998 | 0.037 | 0.816 | 0.043 | 51.06% |
| | TimesNet | 0.783 | 0.929 | 0.910 | 0.992 | 0.039 | 1.001 | 0.044 | 51.03% |
| | DeepSVDDTS | 0.694 | **0.861** | 0.719 | **0.964** | **0.052** | 0.626 | 0.055 | **53.55%** |
| SMD | COUTA | 0.675 | 0.956 | 0.624 | 0.966 | 0.083 | 0.386 | 0.060 | 77.99% |
| | TimesNet | 0.827 | 0.995 | 0.755 | 0.964 | 0.032 | 0.267 | 0.036 | 56.69% |
| | DeepSVDDTS | 0.725 | **0.958** | 0.733 | **0.957** | **0.255** | 2.318 | 0.150 | **93.94%** |
| MSL | COUTA | 0.911 | 0.993 | 0.706 | 0.966 | 0.057 | 0.534 | 0.062 | 55.92% |
| | TimesNet | 0.744 | 0.956 | 0.910 | 0.993 | 0.056 | 0.262 | 0.056 | 62.18% |
| | DeepSVDDTS | 0.825 | **0.973** | 0.816 | **0.979** | **0.123** | 2.347 | 0.139 | **73.89%** |
| NIPS-TS-SWAN | COUTA | 0.780 | 0.788 | 0.738 | 0.710 | 0.022 | 0.574 | 0.064 | 14.52% |
| | TimesNet | 0.770 | 0.906 | 0.773 | 0.866 | 0.013 | 0.451 | 0.080 | 10.56% |
| | DeepSVDDTS | 0.740 | **0.827** | 0.744 | **0.830** | **0.227** | 2.242 | 0.196 | **71.05%** |
| NIPS-TS-CREDITCARD | COUTA | 0.192 | 0.900 | 0.003 | 0.300 | **0.232** | 0.450 | 0.046 | **99.86%** |
| | TimesNet | 0.422 | **0.942** | 0.450 | **0.846** | 0.028 | 0.262 | 0.035 | 51.75% |
| | DeepSVDDTS | 0.132 | 0.772 | 0.119 | 0.719 | 0.105 | 0.817 | 0.056 | 91.55% |
| NIPS-TS-WATER | COUTA | 0.515 | 0.537 | 0.598 | 0.989 | 0.070 | 0.359 | 0.034 | 95.18% |
| | TimesNet | 0.778 | 0.997 | 0.550 | 0.974 | 0.120 | 0.279 | 0.032 | 99.07% |
| | DeepSVDDTS | 0.512 | **0.764** | 0.513 | **0.907** | **0.189** | 0.540 | 0.039 | **99.46%** |
| UCR-1 | COUTA | 0.672 | 0.986 | 0.949 | 0.999 | 0.177 | 0.413 | 0.124 | 69.43% |
| | TimesNet | 0.886 | 0.996 | 0.845 | 0.995 | 0.011 | 0.207 | 0.024 | 25.30% |
| | DeepSVDDTS | 0.813 | **0.994** | 0.984 | **1.000** | **0.351** | 0.758 | 0.220 | **74.89%** |
| UCR-2 | COUTA | 0.886 | 0.998 | 0.842 | 0.939 | 0.022 | 0.190 | 0.033 | 38.84% |
| | TimesNet | 0.984 | **1.000** | 0.982 | **1.000** | **0.036** | 0.256 | 0.042 | **52.15%** |
| | DeepSVDDTS | 0.118 | 0.900 | 0.306 | 0.970 | 0.033 | 0.218 | 0.043 | 46.02% |

Table 1: Detection performance and certified robustness of the DTW-certified defense across various datasets and models with hyperparameter $\sigma = 0.5$. The results show minimal degradation in detection performance while consistently achieving meaningful DTW-certified robustness.

generous attack budget of $e_{att} = 1.0$, which exceeds the average certified radius of $0.5$ measured by both $\ell_p$-norm and proposed DTW-certified defenses across datasets for model COUTA. As shown in Table 2, the DTW-based adversarial attack is highly effective against undefended models, causing significant drops in F1-score and ROC AUC (e.g., a 60.6% drop in F1-score on SMD and 89.7% on UCR-1). While the $\ell_p$-norm certified defense offers partial resilience, it fails to provide consistent protection, especially on datasets with strong temporal distortions under attacks (e.g., SMD and NIPS-TS-WATER). In contrast, our DTW-certified defense consistently outperforms both baselines under attack, yielding substantially higher F1-scores and AUCs. For example, on MSL and UCR-1, our method improves the F1-score under attack by 11.2% and 18.7%, respectively, compared to the $\ell_p$-norm certified defense. These results affirm that robustness guarantees aligned with DTW—rather than $\ell_p$-norm—are essential for effective defense in time-series anomaly detection.

**Trade-off between detection performance and certified robustness.** We investigate the trade-off between detection performance and certified robustness by varying the hyperparameter $\sigma$, which

| Dataset | Unattacked | | Under DTW-based Adversarial Attack with Attack Budget DTW $e_{att} = 1.0$ | | | | | |
|---|---|---|---|---|---|---|---|---|
| | Base Model | | Undefended Base Model | | $\ell_p$-norm Certified Defense | | DTW-Certified Defense | |
| | F1-score | ROC AUC | F1-score | ROC AUC | F1-score | ROC AUC | F1-score | ROC AUC |
| MSL | 0.896 | 0.992 | 0.694 | 0.938 | 0.672 | 0.943 | **0.784** | **0.966** |
| SMD | 0.575 | 0.938 | 0.253 | 0.698 | 0.392 | 0.741 | **0.464** | **0.838** |
| NIPS-TS-WATER | 0.516 | 0.689 | 0.240 | 0.665 | 0.423 | 0.797 | **0.525** | **0.927** |
| UCR-1 | 0.682 | 0.987 | 0.084 | 0.846 | 0.761 | 0.893 | **0.948** | **0.998** |

Table 2: Detection performance under DTW-based adversarial attacks, evaluated using the COUTA model across multiple datasets. Comparisons are made among the undefended base model, the $\ell_p$-norm certified defense, and the proposed DTW-certified defense.

| Dataset | $\sigma$ | DTW-Certified Defense | | | | | |
|---|---|---|---|---|---|---|---|
| | | Detection Performance | | Certified Robustness (Radii Statistics) | | | |
| | | F1-score | ROC AUC | Radii Mean | Radii Max | Radii Std. | Certified Prop. |
| SMAP | 0.1 | 0.956 | 0.997 | 0.007 | 0.320 | 0.010 | 41.13% |
| | 0.5 | 0.961 | 0.998 | 0.037 | 0.816 | 0.043 | 51.06% |
| | 1.0 | **0.961** | **0.998** | 0.080 | 1.051 | 0.082 | 58.02% |
| | 2.0 | 0.913 | 0.958 | **0.202** | 1.571 | 0.165 | **72.24%** |
| SMD | 0.1 | 0.504 | 0.881 | 0.025 | 0.190 | 0.028 | 52.42% |
| | 0.5 | 0.624 | 0.966 | 0.083 | 0.386 | 0.060 | 77.99% |
| | 1.0 | **0.673** | **0.977** | 0.121 | 0.511 | 0.084 | 83.28% |
| | 2.0 | 0.315 | 0.822 | **0.456** | 1.969 | 0.285 | **94.66%** |
| MSL | 0.1 | **0.841** | 0.982 | 0.003 | 0.426 | 0.020 | 11.13% |
| | 0.5 | 0.706 | 0.966 | 0.057 | 0.534 | 0.062 | 55.92% |
| | 1.0 | 0.830 | **0.984** | 0.108 | 0.457 | 0.098 | 68.27% |
| | 2.0 | 0.739 | 0.914 | **0.294** | 1.457 | 0.196 | **87.72%** |
| NIPS-TS-WATER | 0.1 | 0.515 | 0.775 | 0.192 | 0.473 | 0.038 | **99.42%** |
| | 0.5 | **0.598** | **0.989** | 0.070 | 0.359 | 0.034 | 95.18% |
| | 1.0 | 0.555 | 0.986 | 0.161 | 0.430 | 0.061 | 98.79% |
| | 2.0 | 0.462 | 0.983 | **0.261** | 0.711 | 0.114 | 98.54% |
| UCR-1 | 0.1 | 0.871 | 0.996 | 0.023 | 0.112 | 0.027 | 50.63% |
| | 0.5 | **0.949** | **0.999** | 0.177 | 0.413 | 0.124 | 69.43% |
| | 1.0 | 0.919 | 0.984 | 0.309 | 0.692 | 0.196 | 75.36% |
| | 2.0 | 0.821 | 0.973 | **0.541** | 1.209 | 0.300 | **82.67%** |

Table 3: Detection performance and certified robustness results evaluated under varying $\sigma = \{0.1, 0.5, 1.0, 2.0\}$ using COUTA. Higher $\sigma$ generally yield improved certified robustness (Mean, Max, Prop.), but could at the expense of reduced detection performance (F1-socre, ROC AUC).

controls the magnitude of Gaussian noise $\mathcal{N}(0, \sigma^2 I)$ used in the smoothing process. Notably, under a moderate setting ($\sigma = 0.5$), many configurations exhibit improved detection performance compared to the base model, as shown in datasets SMAP and UCR-1 in Table 1. This observation is consistent with prior work [18, 73, 43, 26], where smoothing is shown to enhance generalization by stabilizing decision boundaries. As illustrated in Table 3, increasing $\sigma$ generally improves certified robustness, as reflected in larger average certified radii and a higher proportion of certified inputs. However, overly large values (e.g., $\sigma = 2.0$) often degrade detection performance, including both F1-score and ROC AUC. Therefore, the choice of $\sigma$ should be tuned carefully, considering both the model architecture and dataset characteristics.

# 5 Conclusion and Limitations

We present the first certified defense for time-series anomaly detection under the Dynamic Time Warping (DTW) distance—a metric well-suited for capturing temporal structure in time-series data. By adapting randomized smoothing and leveraging the Keogh lower bound, we derive a DTW-certified radius that provides formal robustness guarantees. This method is model-agnostic across diverse datasets and architectures. Empirical results demonstrate that it consistently delivers strong DTW-certified robustness while maintaining strong detection performance.

The Monte Carlo sampling process introduces testing-time overhead, which future work may address by exploring more efficient sampling strategies or adaptive noise injection methods. Additionally, tightening the DTW relaxation by incorporating more precise lower bounds could lead to stronger robustness guarantees. Finally, extending the proposed framework to broader time-series tasks, such as classification, presents a promising direction for future research.

## Acknowledgements

We thank Dr. Tarun Soni and Kerry Brown for their helpful discussions and valuable feedback. Sarah Monazam Erfani is in part supported by the Australian Research Council (ARC) Discovery Early Career Researcher Award (DECRA) DE220100680.

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

## A  Randomized Smoothing Details

Randomized smoothing [17] was originally proposed and widely applied in classification tasks by constructing the *smoothed function* $g(x)$ that takes the Gaussian means of the base function $f$ as

$$g(x) = \mathbb{E}_{\eta \sim N(0,\sigma^2 I)}[f(x + \eta)] \ . \tag{12}$$

**Lemma A.1.** *Given a bounding output range as $f : \mathcal{X} \to [l, u]$, the upper and lower bounds on the output of the Gaussian smoothed function $g(x')$ can be shown as [14]*

$$l + (u - l) \cdot \Phi\left(\frac{k(x) - \|x - x'\|_2}{\sigma}\right) \ \le \ g(x') \ \le \ l + (u - l) \cdot \Phi\left(\frac{k(x) + \|x - x'\|_2}{\sigma}\right) \ , \tag{13}$$

*where $k(x) = \sigma \cdot \Phi^{-1}(\frac{g(x)-l}{u-l})$ and $\Phi$ denote the cumulative distribution function (CDF) of the standard Gaussian distribution.*

In the context of classification, $g(x)$ is interpreted as the bounded probability score in range $[0, 1]$ for each label, e.g., the softmax score, and a certificate can be obtained by bounding the gap between the highest and second-highest scores.

## B  Proof of Theorem 3.3

Here we provided the complete proof of the Theorem 3.3.

*Proof.* First, we prove that $d(x') = d(x)$ holds for all $x'$ satisfying $\|x - x'\| \le r$. In the case of $d(x) = 1$, i.e., $h_p(x) > \gamma$, we prove that all $x'$ satisfies $h_p(x') > \gamma$. Given the inequality $h_{\underline{p}}(x) < h_p(x')$ for all $\|x - x'\|_2 < r$, as established in Lemma 3.1, it follows that if $h_{\underline{p}}(x) > \gamma$, then $\gamma < h_{\underline{p}}(x) < h_p(x')$, which ensures that $h_p(x') > \gamma$. In the case of $d(x) = 0$, the proof follows by a similar argument. The radius $r$ can be solved by the definitions of $\underline{p} = \Phi(\Phi^{-1}(p) - \frac{r}{\sigma})$ and $\overline{p} = \Phi(\Phi^{-1}(p) + \frac{r}{\sigma})$ as given in Lemma 3.1.

By Lemma 3.2, such certification can be transited to $\{x' \mid DTW(x, x') \le e\}$ by solving the $e = \inf\{LB(x, x') \mid \|x - x'\| > r\}$ with a proper choice of the lower bound $LB(x, x')$.

We consider the Keogh Lower Bound [28] $LB\_Keogh(x, x')$, which is a strict lower bound of $DTW$ for any $w > 0$ and $x' \ne x$. The $LB\_Keogh(x, x')$ is calculated as the sum of deviation of $x'$ outside the envelope of $x$. For each time step $i$ we define the slack (allowable deviation) without incurring any penalty in $LB\_Keogh(x, x')$ by $\Delta_i = \max(U_i - x_i, \ x_i - L_i)$ and the sum of all time steps as $R = \sqrt{\sum_{i=1}^n \Delta_i^2}$. Thus, if $x'$ could "hide" all the norm deviation $r$ within the slacks for all time steps as $r \le R$, the $LB\_Keogh(x, x') = 0$. Hence, in that case, $e = 0$.

Then consider the case when $r > R$, which means any $x'$ with $\|x - x'\| > r$ must have at least one coordinate outside the envelope. Since we are solving for the infimum, the smallest possible $LB\_Keogh(x, x')$ is when $\|x - x'\| = r$ and the $x'$ use up the available slack $\Delta_i$ in every coordinate except one $i^*$ where the slack $\Delta_{i^*}$ is largest. Then, such a worst-case $x'$ is defined as

$$x_i' = \begin{cases} x_i + \Delta_i, & \text{if } i \ne i^*, \\ x_{i^*} + \Delta_{i^*} + d, & \text{if } i = i^*. \end{cases} \tag{14}$$

with $d$ as the part outside the envelops and $\|x - x'\| = r$. In that case

$$r^2 = \|x - x'\|^2 = \sum_{i \ne i^*} \|\Delta_i\|^2 + \|\Delta_{i^*} + d\|^2 = \left(\sum_{i=1}^n \Delta_i^2\right) + 2\, d^T \Delta_{i^*} + \|d\|^2 \tag{15}$$

Note that the $LB\_Keogh(x, x')$ is calculated as the sum of deviations outside the envelope. Thus, the infimum $e$ when $r > R$ can be obtained by solving $\|d\|$. To yield the extreme value of $\|d\|$, $d$ and $\Delta_{i^*}$ should be collinear and can be written as $d = \lambda \Delta_{i^*}$. With the substitution, the equation becomes

$$(\lambda^2 + 2\lambda)M^2 + R^2 = r^2 \ . \tag{16}$$

Solve for the $\lambda$, we have

$$\lambda = -1 \pm \sqrt{1 + \frac{r^2 - R^2}{M^2}} \ . \tag{17}$$

Therefore, the infimum value of $\|d\|$ is

$$\|d\| = \sqrt{M^2 + r^2 - R^2} \ - \ M = e \ . \tag{18}$$

$\square$

## C   Extension to $\ell_p$ Norm

**Generalization of Randomized Smoothing to Arbitrary Norms**   Our certified robustness analysis in the main text is built upon randomized smoothing under the $\ell_2$ norm using Gaussian noise. The framework naturally extends to arbitrary $\ell_p$ norms by replacing the isotropic Gaussian distribution with a noise distribution that is radially symmetric with respect to the chosen norm [72]. Let $\| \cdot \|_p$ denote the base norm and $\| \cdot \|_q$ its dual norm, where $\frac{1}{p} + \frac{1}{q} = 1$. We consider a noise vector $\eta$ drawn from a distribution that is *spherically symmetric* with respect to $\| \cdot \|_p$, such that the density of $\eta$ depends only on $\|\eta\|_p/s$, where $s$ is a scale parameter. Typical choices include: $\ell_2$ with Gaussian noise $\eta \sim \mathcal{N}(0, \sigma^2 I)$; $\ell_1$ with Laplace noise $\eta_i \sim \text{Laplace}(0, b)$ i.i.d.; $\ell_\infty$ with Uniform noise $\eta_i \sim \text{Unif}[-\tau, \tau]$ i.i.d. General $\ell_p$ can use generalized Gaussian noise with density proportional to $\exp(-\|\eta\|_p^\alpha/\lambda^\alpha)$ for $\alpha = p$.

**Quantile Stability under $\ell_p$ Perturbations**   Let $f$ denote the base anomaly scoring function and $h_p(x)$ the $p$-th percentile of $f(x + \eta)$ under the smoothing noise distribution. For any $r \geq 0$, define $F$ as the cumulative distribution function of $\langle u, \eta \rangle$ for any unit vector $u$ with $\|u\|_q = 1$. Then, the following holds for all $\|x' - x\|_p \leq r$:

$$h_{F(F^{-1}(p)-r/s)}(x') \leq h_p(x) \leq h_{F(F^{-1}(p)+r/s)}(x'). \tag{19}$$

Equation (19) generalizes the Gaussian case by replacing the standard normal CDF $\Phi$ with the 1-D marginal $F$ of the chosen noise distribution, and the Gaussian scale $\sigma$ with the corresponding scale parameter $s$. This yields a certified radius in $\ell_p$ norm space.

**DTW Certification via $\ell_p$ Lower Bounds**   The DTW-based certification derived in Lemma 3.2 remains valid once $\ell_2$ is replaced by $\ell_p$. Specifically, let $\text{LB}_{\text{Keogh},p}(x, x')$ denote a *strict* lower bound of $\text{DTW}_p(x, x')$. For any perturbation $\|x' - x\|_p \leq r$, the certified DTW radius is

$$e_p = \inf\{ \text{LB}_{\text{Keogh},p}(x, x') : \ \|x' - x\|_p > r \}. \tag{20}$$

A practical closed-form lower bound can be obtained via

$$e_p \ \geq \ \left(r^p - R_{\text{in},p}^p\right)_+^{1/p}, \tag{21}$$

where $R_{\text{in},p}$ is the largest $\ell_p$ ball centered at $x$ fully contained within the envelope $[L, U]$ used in the $\text{LB}_{\text{Keogh},p}$ construction. This provides a conservative yet efficient computation of certified DTW radius for arbitrary norms.

This extension preserves the overall structure of the certification pipeline:

1. Replace Gaussian noise with a norm-symmetric distribution;
2. Replace the Gaussian CDF $\Phi$ by the 1-D marginal CDF $F$ of that noise;
3. Compute the $\ell_p$ certified radius $r$ via (19);
4. Translate the $\ell_p$ certificate into DTW certificate $e_p$ using (20).

This demonstrates that the proposed percentile-based randomized smoothing framework is inherently norm-agnostic, supporting robustness certification under any $\ell_p$ metric and its induced DTW variants.

## D   Dataset Details

- The Soil Moisture Active Passive (SMAP) dataset [48] contains soil moisture and telemetry measurements collected by NASA's Mars rover.

| Datasets | Channels | Training Timesteps | Testing Timesteps | Testing Anomalies Ratio % |
|----------|----------|--------------------|--------------------|---------------------------|
| SMAP | 25 | 135,183 | 427,617 | 13.13% |
| MSL | 55 | 58,317 | 73,729 | 10.72% |
| SMD | 25 | 708,405 | 708,420 | 4.16% |
| NIPS-TS-SWAN | 38 | 60,000 | 60,000 | 32.60% |
| NIPS-TS-CREDITCARD | 29 | 284,807 | 284,807 | 0.17% |
| NIPS-TS-WATER | 9 | 69,260 | 69,260 | 1.05% |
| UCR-1 | 1 | 35,000 | 44,795 | 1.38% |
| UCR-2 | 1 | 35,000 | 45,000 | 0.67% |

Table 4: Statistics of the benchmark datasets for time-series anomaly detection.

- The Mars Science Laboratory (MSL) dataset [27] includes comprehensive sensor and actuator data directly obtained from the Mars rover.
- The Server Machine Dataset (SMD)[60] offers stacked resource utilization data from 28 machines within a compute cluster, collected over a five-week duration.
- The NIPS-TS benchmark suite[30] and the UCR collection [68], which provide standardized datasets widely employed in time-series anomaly detection.

# E Certified Confusion Matrix

| | Predicted Positive | Predicted Negative |
|---|---|---|
| Positive | Certified TP$(t) = \sum_{i=1}^{N} \mathbb{I}\{\forall x' : DTW(x_i, x') \leq t : f(x') = 1, y_i = 1\}$ | $\sum_{i=1}^{N} \mathbb{I}\{y_i = 1\} -$ Certified TP$(t)$ |
| Negative | FP | TN |

Table 5: Certified Confusion Matrix for evasion attacks.

| | Predicted Positive | Predicted Negative |
|---|---|---|
| Positive | TP | FN |
| Negative | $\sum_{i=1}^{N} \mathbb{I}\{y_i = 0\} -$ Certified TN$(t)$ | Certified TN$(t) = \sum_{i=1}^{N} \mathbb{I}\{\forall x' : DTW(x_i, x') \leq t : f(x') = 0, y_i = 0\}$ |

Table 6: Certified Confusion Matrix for availability attacks.

Certified accuracy is a metric widely used in certified robust machine learning, measuring the fraction of examples for which a model can provably maintain correct predictions under specific perturbations. For a certified radius $e$, it is defined as

$$\text{Certified Accuracy}(e) = \frac{1}{N} \sum_{i=1}^{N} \mathbb{I}\{\forall x' \text{ with } DTW(x_i, x') \leq e : f(x') = y_i\} \tag{22}$$

Following the definition of certified accuracy, we construct the certified confusion matrix as described in Section 4. Given the certified confusion matrix, the *certified accuracy* is computed as the proportion of instances for which the model guarantees correct predictions within a perturbation threshold. It is defined as:

$$\text{Certified Accuracy} = \frac{\text{Certified TP} + \text{Certified TN}}{N}, \tag{23}$$

where $N$ is the total number of test instances.

Similarly, the *certified F1-score*, which balances precision and recall under certification constraints, is calculated as:

$$\text{Certified F1} = \frac{2 \cdot \text{Certified Precision} \cdot \text{Certified Recall}}{\text{Certified Precision} + \text{Certified Recall}}, \tag{24}$$

where

$$\text{Certified Precision} = \frac{\text{Certified TP}}{\text{Certified TP} + \text{FP}}, \tag{25}$$

$$\text{Certified Recall} = \frac{\text{Certified TP}}{\text{Certified TP} + \text{FN}}. \tag{26}$$

Here, FP and FN refer to false positives and false negatives, respectively, counted as the remaining instances not included in the certified true predictions. These metrics provide a conservative evaluation of model robustness under worst-case adversarial perturbations.

# F Additional Experiment Results

| Seq. Length $T$ | Window Size $w$ | Standard | | DTW-Certified Defense | | | | | |
|---|---|---|---|---|---|---|---|---|---|
| | | F1-score | ROC AUC | F1-score | ROC AUC | Radii Mean | Radii Max | Radii Std. | Certified Prop. |
| 10 | 2 | 0.571 | 0.856 | 0.671 | 0.943 | 0.088 | 0.337 | 0.053 | 94.40% |
| | 4 | | | | | 0.085 | 0.333 | 0.053 | 92.59% |
| | 10 | | | | | 0.083 | 0.326 | 0.054 | 91.01% |
| 50 | 2 | 0.675 | 0.956 | 0.624 | 0.966 | 0.090 | 0.401 | 0.058 | 82.75% |
| | 4 | | | | | 0.083 | 0.386 | 0.060 | 77.99% |
| | 10 | | | | | 0.074 | 0.374 | 0.061 | 71.05% |
| 100 | 2 | 0.656 | 0.929 | 0.447 | 0.878 | 0.131 | 0.699 | 0.104 | 79.69% |
| | 4 | | | | | 0.119 | 0.678 | 0.107 | 71.54% |
| | 10 | | | | | 0.107 | 0.635 | 0.107 | 63.30% |
| 200 | 2 | 0.681 | 0.963 | 0.440 | 0.880 | 0.057 | 0.392 | 0.076 | 48.50% |
| | 4 | | | | | 0.050 | 0.391 | 0.074 | 41.05% |
| | 10 | | | | | 0.041 | 0.389 | 0.070 | 33.33% |

Table 7: Empirical and certified robustness results for the SMD dataset using the COUTA model with $\sigma = 0.5$, evaluated under varying sequence length $T$ and DTW wrapping window size $w$.

Table 7 presents an ablation study on the impact of sequence length $T$ and DTW wrapping window size $w$ on both detection performance and certified robustness. The results indicate a trade-off between these parameters and robustness guarantees. Increasing the sequence length generally enhances detection performance (F1-score and ROC AUC) by incorporating more temporal context for anomaly detection. However, this comes at the cost of reduced certified radius, as the higher dimensionality magnifies the impact of injected noise. Similarly, increasing the wrapping window $w$ allows greater temporal flexibility in DTW alignment but leads to looser Keogh lower bounds and higher slack (as defined by the value $R$ in Theorem 3.3), thereby weakening the robustness guarantee.

# G Border Impact

Time-series anomaly detection plays a crucial role in many safety-critical domains, including healthcare monitoring, financial fraud detection, industrial control systems, and mobile communication networks. In such applications, robustness to adversarial manipulation is not only a matter of performance but also of safety, reliability, and trust. This work contributes to the broader goal of deploying machine learning systems that are resilient to worst-case perturbations in time-series data, particularly those involving temporal distortions.

Our proposed DTW-certified defense offers a principled approach to formally quantifying and improving the robustness of anomaly detection systems under realistic threat models. By aligning the certification metric with the temporal structure of time-series data, we aim to enable more reliable AI systems in high-stakes environments. However, we acknowledge that any advancement in robustness may also encourage the development of stronger adversarial strategies. As such, we encourage responsible deployment and continuous evaluation of these defenses in real-world conditions.

This work is primarily beneficial to organizations seeking reliable time-series analytics in critical domains. It does not disproportionately disadvantage any particular group. Nonetheless, as with any security-related research, care should be taken to ensure that the methodology is not misused to benchmark or strengthen attack strategies without accompanying safeguards.

