# OpenReview forum: "Fortifying Time Series: DTW-Certified Robust Anomaly Detection"
_NeurIPS.cc/2025/Conference — NeurIPS 2025 poster_

### Official Review · Reviewer_XuxL · 2025-06-30

**Clarity:** 3
**Significance:** 3
**Originality:** 2
**Rating:** 4
**Confidence:** 5

**Summary:**

The paper proposes a novel DTW-certified robustness framework by adapting randomized smoothing approach. The proposed method bridges l-norm to DTW distances by a lower-bound transformation. Extensive experiments on various baselines and datasets are conducted to showcase the advantages of the proposed method.

**Questions:**

Q1. What’s the time cost of the proposed method? (W1)
Q2. Can the proposed method scale to million-scale datasets? (W2)
Q3. What are the impacts of different number of noisy samples and percentile p? (W3)

**Ethical Concerns:**

["NO or VERY MINOR ethics concerns only"]

**Limitations:**

yes

**Paper Formatting Concerns:**

no major issues

**Quality:**

3

**Strengths And Weaknesses:**

Strengths:
S1. The paper provides a novel theoretical framework for bridging the l-norm and DTW-distances.
S2. The paper proposes a robust and general defense mechanism that is model-agnostic.
S3. The paper includes extensive experiments on various baselines and datasets.
Weaknesses:
W1. Lacks efficiency studies. No time cost analysis or empirical studies.
W2. Lacks large scale datasets, i.e., million-scale.
W3. Lacks more comprehensive hyperparameter studies, e.g. studies on number of noisy samples and percentile p.
W4. Figure 1 could be improved with legends to indicates the adversarial changes.

---

> ### Author Rebuttal · Authors · 2025-07-31
>
> We sincerely thank the reviewer for the valuable feedback provided. Below, we clarify and address each concern in detail:
>
> **Q1** As shown in Figure 2, the DTW-based certification process uses an analytical solution that introduces no computational overhead. The observed slowdown arises from the Monte Carlo sampling step, which increases inference time roughly in proportion to the number of noisy samples $n$ (e.g., $n$=1000). The parameter $n$ is user-configurable: larger values yield more precise estimates and typically result in a larger certified radius.
>
> We would like to note that such overhead is intrinsic to all randomized smoothing methods [32][18][55], reflecting a fundamental trade-off for significantly stronger robustness guarantees. Nevertheless, randomized smoothing is one of the most efficient and scalable approaches compared to other certified robustness methods [18]. Our method does not incur any additional cost beyond that of standard randomized smoothing, and reducing sample complexity remains an active research direction. In practice, the overhead can be substantially reduced through parallelization using multiple copies of the anomaly detector.
>
> We provide additional results on running time overhead, evaluated in the settings as described in Section 4.
> * Normal inference: 1/498s per batch (size 64)
> * Inference and Certification: 1.83s per batch (size 64)
>
> **Q2** Our method does not require any modification to the training procedure and therefore imposes no computational overhead during training.
>
> Furthermore, we believe our approach can scale to million-scale testing datasets. As evidence, the SMD dataset evaluated in our experiments contains approximately $0.7$ million ($7 x 10^5$) testing timesteps with $25$ channels, and required $3:36:19$ (h:m:s) to complete both inference and certification. The computational overhead at testing time grows linearly with the dataset size, maintaining an inference time complexity of $O(n)$, equivalent to that of a standard model.
>
> **Q3** The impact of noisy samples is addressed in Q1. The percentile parameter $p$ has little impact on results as long as it reflects the majority distribution and is not overly influenced by outliers (i.e., extreme values $p = 0.0$ or $p = 1.0$). We will include a more comprehensive ablation study in the camera-ready version to further examine its effect.
>
> We sincerely thank the reviewer again for your time and effort. We hope that our responses have clarified the issues raised.

---

> > ### Comment · Reviewer_XuxL · 2025-08-04
> >
> > Thanks for the explanation. I will keep the score.

---

### Official Review · Reviewer_JBUS · 2025-07-03

**Clarity:** 3
**Significance:** 3
**Originality:** 3
**Rating:** 4
**Confidence:** 3

**Summary:**

Existing defenses for adversarial samples are either heuristic or only provide certified-robustness under l_p-norm. Time-series data has a unique characteristic where a small temporal shift may result in a much larger l_p-norm distance. This paper integrates DTW (dynamic time warping) similarity metric with l_p-norm guarantee, and proposes the first DTW-certified robust defense in time-series anomaly detection.

**Questions:**

How would an attack evolve to bypass this DTW-certified mechanism?

In general how big is the test-time overhead compared to baseline methods? Would it be a concern for adoption?

**Ethical Concerns:**

["NO or VERY MINOR ethics concerns only"]

**Limitations:**

yes

**Quality:**

3

**Strengths And Weaknesses:**

Strength:

This paper addresses a meaningful problem where the existing l_p-norm fails to preserve semantic meaning in temporal data, and is the first work to do so.

Theoretical proofs are rigorous and solid.

Evaluation results are comprehensive and demonstrate model-agnostic applicability.

Weakness:

Although computation overhead by monte carlo approach is mentioned as a known limitation, including a comparison on runtime would give a clue on how big this overhead can be.

The discussion of various attacks that may or may not bypass the proposed defense is also helpful for potential users to decide whether this is applicable to their system.

---

> ### Author Rebuttal · Authors · 2025-07-31
>
> We sincerely thank the reviewer for the valuable feedback provided. Below, we clarify and address each concern in detail:
>
> **Computation overhead:** The Monte Carlo sampling introduces computational overhead, slowing inference by approximately a factor of $n$ (the number of noisy samples). The value of $n$ can be user-specified, where a larger $n$ gives a more precise estimation and usually a larger certified radius.
>
> We would like to note that such overhead is intrinsic to all randomized smoothing methods [32][18][55], reflecting a fundamental trade-off for significantly stronger robustness guarantees. Nevertheless, randomized smoothing is still one of the most efficient and scalable approaches compared to other certified robustness methods [18].
>
> Our method does not incur any additional cost beyond that of standard randomized smoothing, and reducing sample complexity remains an active research direction. In practice, the overhead can be substantially reduced through parallelization using multiple copies of the anomaly detector.
>
> We provide additional results on running time overhead, evaluated in the settings as described in Section 4.
> * Normal inference: 1/498s per batch (size 64)
> * Inference and Certification: 1.83s per batch (size 64)
>
>
> **Attack to bypass this defence:**
> As a certified defense method, we theoretically guarantee that no adversarial attack can bypass our approach as long as the adversarial example x' lies within the certified DTW radius e, i.e., DTW(x, x') < e (see Section 3.1, Threat Model and Certified Defense Goal). An attacker can only compromise the defense by introducing a perturbation exceeding the certified radius.
>
> We sincerely thank the reviewer again for your time and effort. We hope that our responses have clarified the issues raised.

---

> > ### Author Response · Authors · 2025-08-05
> >
> > We sincerely appreciate the reviewer’s time and effort, and we hope our rebuttal has addressed the concerns raised. Please let us know if there are any remaining questions or if further clarification would be helpful.

---

### Official Review · Reviewer_CMCZ · 2025-07-03

**Clarity:** 2
**Significance:** 2
**Originality:** 2
**Rating:** 4
**Confidence:** 4

**Summary:**

This paper aims to provide provable robustness for time series anomaly detection under Dynamic Time Warping (DTW) distance. The authors argue that existing Lp-norm based certification methods are unsuitable for time series data due to their sensitivity to temporal distortions. Building upon this observation, they propose a novel certification approach that bridges Lp-norm to DTW distance through randomized smoothing and lower-bound transformation. The proposed method is model-agnostic and has been experimentally validated across multiple datasets.

**Questions:**

See my comments on the weakness.

**Ethical Concerns:**

["NO or VERY MINOR ethics concerns only"]

**Final Justification:**

My comments are well-addressed , and I would like to raise my score.

**Limitations:**

While the paper introduces a theoretically sound DTW-based certification approach with broad applicability, several weaknesses limit its impact. The comparison with Lp-norm certification is arguably unfair, as DTW-specific attacks inherently favor DTW metrics. Practical effectiveness is questionable, with extremely small verifiable radii (e.g., 0.022) and low certified proportions on key datasets (SMAP, NIPS-TS-SWAN), revealing vulnerability to high-dimensional, high-variance data. The method’s novelty is unclear—it combines randomized smoothing and DTW without sufficiently differentiating from prior work (e.g., median smoothing in [14]) or justifying design choices (e.g., exclusive use of sup function). Presentation issues (missing citations, unclear notation) and weakly articulated contributions further undermine the work. Most critically, experiments lack validation across diverse methods, and performance gains are inconsistent, leaving the method’s added value unconvincing.

**Quality:**

2

**Strengths And Weaknesses:**

Strengths:
1. The research problem addressed in this paper is meaningful. Identifying more appropriate robustness metrics (e.g., DTW) for time series data compared to conventional Lp-norm measures represents a valuable research direction.
2. The model-agnostic nature of the proposed methodology substantially enhances its potential applicability across diverse domains.
3. The authors propose a novel certification approach that bridges Lp-norm to DTW distance through randomized smoothing and lower-bound transformation, supported by a strong theoretical framework.

Weaknesses：
1. Although the paper compares with Lp-norm certification, this comparison may not be entirely fair. The DTW attack itself is specifically designed to maximize DTW distance, so it's expected that Lp-norm certification would perform poorly under such attacks.
2. The results (Table 1) show that on some data sets (such as SMAP, NIPS-TS-SWAN), the mean of the verifiable radius is very small (such as 0.037,0.022), and the certified proportion is not high. This shows that the actual protection ability of the method is very limited on these datasets. Although the explanation (high dimension and large data variance) of this paper is reasonable, it also exposes the vulnerability of the method.
3. What is the difference between the smoothed anomaly score function and the median smoothing in [14]? Why only the sup function is used to represent this, but no inf function?
4. A literature citation is required when Dynamic Time Warping first appears in the paper.
5. X= RL×C should be change to XRL×C
6. The contributions of abstact and introduction are weak. In the method part, this work simply combines random smoothing and DTW technology, and the contribution of the statement is not fully reflected in the method part. In the experimental part, the feasibility of the theory is not verified in more methods, and the performance improvement brought by the method is not uniform。

---

> ### Author Rebuttal · Authors · 2025-07-31
>
> We sincerely thank the reviewer for the valuable feedback provided. Below, we clarify and address each concern in detail:
>
> **Concern in Weakness 1 "Comparison with $l_p$-Norm Defenses":** As discussed in the Abstract and Introduction (lines 30-49), the inadequacy of existing $l_p$-norm certification methods for time-series data, their poor performance under DTW-based attacks, and the complete absence of any prior certified defense against DTW attacks were the primary motivations for developing our proposed DTW-certified defense. The performance gap between $l_p$-norm certification and our DTW-certified defense validates our assumption and clearly demonstrates the superiority of the proposed approach under realistic DTW-based adversarial scenarios.
>
> **Concern in Weakness 2 "Performance Differences Across Datasets":** We acknowledge that our certified defense does not achieve identical performance across all datasets. However, it is important to note that no single method/setting achieves optimal performance for all data, and this is a well-known phenomenon in existing certified robustness literature [32][18][55].
>
> As discussed in lines 265-267 & 292-298, the variation arises from differences in dataset characteristics and model architectures, particularly their sensitivity to injected noise. Consequently, tuning the noise level is often necessary to achieve optimal certification performance in each setting.
> For example, on SMAP (COUTA), the mean certified radius is $0.037$ with a certified proportion of $51.06$% at noise level $0.5$ (Table 1), but improves to a mean radius of $0.202$ and certified proportion of $72.24$% at noise level $2.0$ (Table 2).
>
> Since the primary focus of this work is to introduce a novel DTW-based certified defense, we deliberately reported results at a fixed noise level (Table 1) rather than cherry-picking and tuning hyperparameters for each dataset. Table 2 further demonstrates that, with minor tuning, our method can achieve consistently high certified radii and proportions in most settings.
>
>
> **Concern in Weekness 3 & 6 "Contribution and Difference with [14]":** We agree with the first point in *Strengths* that our main contribution lies in providing a theorem for a more appropriate certified robustness metric, DTW, as formalized in Lemma 3.2 and Theorem 3.3 (Section 3.2). In contrast, the method in [14] applies exclusively to $l_p$-norm metrics.
>
> While we incorporate median smoothing from [14] as part of our anomaly score construction, the certification theorem we propose for DTW is non-trivial. It is the first in the literature to provide a certified guarantee in DTW distance. Importantly, a DTW radius generally encompasses more adversarial examples than an $l_p$-norm radius of the same size, meaning that our DTW certificate cannot be derived through a simple adaptation or trivial combination of existing $l_p$-norm results with DTW definition.
>
> **Concern in Weakness 3 "no inf function":** In applying median smoothing, we note that the $\sup$ and $\inf$ functions in [14] Definition 1 are equivalent for continuous distributions, as stated in [14] *“The inf and sup are equivalent for continuous distributions; the distinction is needed to handle edge cases with discrete distributions.”*
>
> **Concern in Weakness 4 & 5 "Notation and Citations":** We will add the corresponding citations and update the notation in the camera-ready version.
>
> **Concern in Limitation "Experimental Validation Across Diverse Methods":** We evaluated our method on three representative state-of-the-art (SOTA) anomaly detection models across eight widely used time-series anomaly detection datasets as reported in Section 4. The experimental setup is consistent with best practices in both the time-series anomaly detection [4][58][59] and certified robustness [18][55] communities.
>
> We sincerely thank the reviewer again for your time and effort. We hope that our responses have clarified the issues raised and kindly request your consideration.

---

> > ### Author Response · Authors · 2025-08-05
> >
> > We sincerely appreciate the reviewer’s time and effort, and we hope our rebuttal has addressed the concerns raised. Please let us know if there are any remaining questions or if further clarification would be helpful.

---

### Official Review · Reviewer_7Njo · 2025-07-04

**Clarity:** 3
**Significance:** 3
**Originality:** 3
**Rating:** 4
**Confidence:** 3

**Summary:**

The paper introduces the first certified‐robust defense for time-series anomaly detection measured in Dynamic Time Warping (DTW) distance. It adapts randomized smoothing to obtain an ℓ₂-norm certificate and then analytically translates this guarantee to DTW by exploiting the Keogh lower bound, yielding a closed-form certified radius that can be reported at inference time. Because the smoothing wrapper sits on top of any pre-trained detector, the method is model-agnostic and requires no retraining. Empirical tests on seven benchmark datasets and three modern detectors show that the approach maintains high detection accuracy while improving F1 scores against strong DTW adversaries, substantiating both its practicality and theoretical soundness.

**Questions:**

1. Generality Beyond Euclidean Norms.
The paper states that the proposed certificate “readily generalizes to arbitrary norm orders p” (line 125). However, both the theoretical development and the experiments appear tailored to the Euclidean norm. Could the authors provide a concrete derivation for at least one non-Euclidean case (e.g., p=1) and report certified radii on the same datasets?

2. Smoothness Assumption in Lemma 3.1.
Lemma 3.1 hinges on smooth anomaly scores, yet the manuscript does not explicitly state any regularity conditions on the anomaly score function $f$. I encourage the authors to formally state the smoothness or continuity assumptions required for the lemma to hold, and possibly include a brief proof sketch showing that the percentile smoothing preserves these properties.

3. Inference Efficiency and Latency.
The proposed method relies on large numbers of Monte Carlo samples (e.g., $n = 1000$ in experiments), which may cause considerable inference latency. It would be helpful if the authors could include measurements of wall-clock latency or throughput, particularly for typical values of $n$.

**Ethical Concerns:**

["NO or VERY MINOR ethics concerns only"]

**Final Justification:**

My concerns have been thoroughly addressed through the rebuttal. I have updated my score accordingly.

**Limitations:**

Yes

**Quality:**

3

**Strengths And Weaknesses:**

**Strengths**

1. The paper closes a gap by deriving a formal certificate in Dynamic Time Warping (DTW) distance rather than the usual $l_p$ norms, which are known to mis-characterise temporal distortions. This is positioned as the first such result in this domain.

2. Because the defense is applied post-training (it wraps a pre-trained detector with a smoothing/denoising layer), practitioners can add robustness onto existing systems without retraining.

3. By proving Lemma 3.2 and Theorem 3.3, the authors map any $l_2$-certified radius obtained via randomized smoothing to an explicit DTW radius through the Keogh lower bound. This connection is practically computable.

**Weaknesses**

1. The paper's robustness guarantee quietly depends on the anomaly-score function changing smoothly whenever the input is nudged a tiny bit. The key lemma assumes that adding a small amount of Gaussian noise cannot reorder which inputs look more or less anomalous, something that holds only when the score has no sudden jumps. The three neural network detectors tested in the experiments behave this way, so the proof is valid for them. But any detector that uses hard thresholds, coarse rounding, or aggressive max-pool operations could let its score flip abruptly under minuscule input changes, breaking the guarantee. Therefore, the defense is provably safe only for detectors that produce smooth, gently varying scores, which narrows its general usefulness.

2. The paper's promise that its method can be extended to any distance norm "with minimal modification" is not backed up by the proofs. Every step of the theoretical argument is built around ordinary Euclidean distance: the randomized-smoothing lemma protects the model only against perturbations measured in that specific way, and the later conversion from Euclidean radius to a Dynamic-Time-Warping radius depends on algebra and geometric facts that hold only under Euclidean geometry. The authors never supply a new noise-adding scheme, inequality, or optimization path that would make the same guarantee work for other norms, so the statement in lines 124-126 that such a change would be easy remains an unproven claim.

3. Certified prediction relies on huge amounts of Monte-Carlo samples, adding tangible latency that could hinder real-time monitoring.

---

> ### Author Rebuttal · Authors · 2025-07-31
>
> We sincerely thank the reviewer for the valuable feedback provided. Below, we clarify and address each concern in detail:
>
> **1 Generality to Arbitrary Norm Orders $p$:** Our method builds upon Randomized Smoothing, which has been generalized to arbitrary norm orders $p$ in the cited work \[73]. In summary, different norm orders can be achieved by adding appropriate noise distributions:
> * Gaussian noise for $p = 2$
> * Laplace noise for $p = 1$
> * Uniform noise for $p = \infty$
>
> The proposed DTW-certificate can likewise be generalized to any norm order $p$ (e.g., $p = 1$) by replacing Gaussian noise with the corresponding distribution (e.g., Laplace noise). Regarding the certification process, Lemma 3.2 and its proof remain valid for arbitrary $p$, and Theorem 3.3 requires the following modifications:
>
> 1. Redefine the certified radius $r$ from norm-2 to norm-1.
> 2. Redefine $R$ from norm-2 to norm-1 as $R = \sum \|\delta_i\|$.
> 3. In Eq. (15), replace the norm-2 expansion with norm-1 expansion: $r = \sum \|\delta_i\| + \|d\|$.
> 4. Following the same proof steps, the expression of $e$ in Eq. (10) becomes $e = r - R$.
>
> Given the limited time, we provide initial results for the norm-1 certification of the COUTA model here (align to Table 1, last 6 columns), and will expand on this discussion in greater detail in Sec 4 of the camera-ready.
>
> | Dataset             | F1-score | ROC AUC | Radii Mean | Radii Max | Radii Std. | Certified Prop. |
> |---------------------|----------|---------|------------|-----------|------------|-----------------|
> | SMAP                | 0.959    | 0.998   | 0.058      | 0.444     | 0.078      | 47.23%          |
> | SMD                 | 0.574    | 0.937   | 0.028      | 0.207     | 0.041      | 62.83%          |
> | MSL                 | 0.818    | 0.970   | 0.027      | 0.496     | 0.075      | 44.23%          |
> | NIPS-TS-SWAN        | 0.738    | 0.795   | 0.012      | 0.432     | 0.032      | 12.97%          |
> | NIPS-TS-CREDITCARD  | 0.652    | 0.919   | 0.054      | 0.322     | 0.073      | 74.34%          |
> | NISP-TS-WATER       | 0.445    | 0.969   | 0.083      | 0.217     | 0.021      | 84.36%          |
> | UCR-1               | 0.974    | 0.999   | 0.040      | 0.300     | 0.018      | 57.25%          |
> | UCR-2               | 0.688    | 0.972   | 0.013      | 0.094     | 0.079      | 19.47%          |
>
> **2 Applicability to Arbitrary Anomaly Score Functions:** Our method and Lemma 3.1 impose no smoothness or continuity assumptions on the anomaly score function $f: \mathcal{X} \to \mathbb{R}$. The proposed approach is applicable to any anomaly score function that is used as described in Section 3.1 (Para. Time-series Anomaly Detector), where the anomaly is identified as an anomaly score that exceeds the threshold. This includes the function output with sudden jumps, such as hard thresholds, coarse rounding, or aggressive max-pool operations.
> This generality follows from our use of a randomized smoothing approach, which can be applied to any function as one of its key advantages, as established in [17, 14].
>
> **3 Computational Overhead:** We acknowledge that Monte Carlo sampling introduces computational overhead, slowing inference by approximately a factor of $n$ (the number of noisy samples). The value of $n$ can be user-specified, where a larger $n$ gives a more precise estimation and usually a larger certified radius.
>
> We would like to note that such overhead is intrinsic to all randomized smoothing methods [32][18][55], reflecting a fundamental trade-off for significantly stronger robustness guarantees. Nevertheless, randomized smoothing is still one of the most efficient and scalable approaches compared to other certified robustness methods [18].
>
> Our method does not incur any additional cost beyond that of standard randomized smoothing, and reducing sample complexity remains an active research direction. In practice, the overhead can be substantially reduced through parallelization using multiple copies of the anomaly detector.
>
> We provide additional results on running time overhead, evaluated in the settings as described in Section 4.
> * Normal inference: 1/498s per batch (size 64)
> * Inference and Certification: 1.83s per batch (size 64)
>
> We sincerely thank the reviewer again for your time and effort. We hope that our responses have clarified the issues raised and kindly request your consideration.

---

> > ### Author Response · Authors · 2025-08-05
> >
> > We sincerely appreciate the reviewer’s time and effort, and we hope our rebuttal has addressed the concerns raised. Please let us know if there are any remaining questions or if further clarification would be helpful.

---

> > ### Comment · Reviewer_7Njo · 2025-08-05
> >
> > Thank you for the detailed response. My concerns regarding generalization to arbitrary norm orders, applicability to anomaly score functions, and computational overhead have been addressed. The theoretical clarifications and additional results are convincing.
> >
> > I will raise my score and recommend incorporating these insights into the final version to further strengthen the paper.

---

### Note · Authors · 2025-08-12

Dear AC,

We thank you and all reviewers for the thorough evaluations and constructive feedback. Across the reviews, there is a clear consensus that our work addresses an important problem in certified robustness for time-series anomaly detection, introducing the first DTW-based certified defense with strong theoretical guarantees and model-agnostic applicability. Extensive experiments across diverse datasets and detectors validate the method’s effectiveness under both theoretical guarantee and strong DTW-based attacks.

All reviewer concerns (generality beyond $l_2$, computational efficiency, and methodological clarifications) were addressed in detail, with concrete derivations, empirical results, and camera-ready commitments. Reviewers explicitly confirmed that these points were resolved, and no issues remain outstanding. To our knowledge, the discussion ended with no remaining objections and a clear consensus on the work’s soundness and contribution.

We appreciate the constructive feedback and believe the final version makes a substantial contribution to advancing certified robustness in time-series analysis.

---

### Decision · Program_Chairs · 2025-09-17

**Decision:**

Accept (poster)

**Comment:**

This paper introduces a DTW-based certified defense for time-series anomaly detection, offering a solid theoretical foundation and demonstrating model-agnostic applicability across diverse datasets. Its novelty and rigor make it a valuable contribution to the field. The main limitations concern the reliance on smoothness assumptions, computational overhead, and relatively limited evaluation scope, which constrain its demonstrated practicality. Nonetheless, all reviewers gave positive feedback, and the rebuttal effectively addressed most concerns. Overall, I recommend acceptance, with the expectation that the authors incorporate reviewer suggestions in the final version to strengthen clarity, efficiency analysis, and broader validation.